# Gym4ReaL: Towards Real-World Reference Environments for Reinforcement Learning

## Abstract

In recent years, *Reinforcement Learning* (RL) has achieved remarkable progress, reaching superhuman performance across a variety of simulated environments, largely driven by the adoption of standardized training suites, such as Gymnasium and MuJoCo. However, this success has not been translated directly to real-world domains, which present inherent challenges that remain underexplored in existing reference environments. This gap highlights the need for training suites that more closely reflect real-world conditions and facilitate the practical deployment of RL solutions. Towards this goal, in this paper, we introduce `Gym4ReaL`, an open-source suite of realistic environments developed starting from collaborations with industry partners and domain experts. The suite offers a diverse collection of tasks, simulators, and datasets that expose algorithms to real-world complexities and support the investigation of different methodological approaches. Through benchmark experiments, we demonstrate that standard RL algorithms remain competitive against expert-guided rule-based baselines in these settings, motivating the development of new methods capable of fully harnessing RL's potential for real-world applications.

## 1 Introduction

In the past few years, *Reinforcement Learning* (RL) (Sutton & Barto, 2018) has demonstrated above-human performance across different challenges, ranging from playing Atari games (Mnih et al., 2015) to beating world champions of Chess and Go (Silver et al., 2017a;b), achieving impressive results also in the field of robotic control (Kober et al., 2013). However, despite these promising advances, RL still struggles to gain traction in many real-world applications, where systems are often subject to uncertainties and unpredictable factors that complicate physical modeling. An additional limitation lies in the fact that RL algorithms are typically validated on idealized environments, such as those provided by Gymnasium (Towers et al., 2024) and MuJoCo (Todorov et al., 2012). Despite their great contribution to RL research, such libraries provide artificial playgrounds able to generate infinite samples, adapt to any desired configuration, and grant harmless exploration. However, learning and overfitting these environments does not necessarily reflect skillfulness in real-world tasks, where data availability is limited, dynamics change, and exploration does not come for free.

From this perspective, the collaboration with industry and domain experts – who can provide operational objectives, validated simulators, and real datasets – may contribute meaningfully to RL research. Our work takes a first, practical step in this direction, toward narrowing the gap between theoretical RL analyses and realistic operational settings, aiming to promote techniques with demonstrable applicability. We therefore present `Gym4ReaL`, a reference environment suite developed in collaborations with industries and research centers and designed to realistically model several real-world environments under a unified interface, grounded in research-grade simulators and real-world datasets. The selected tasks included in `Gym4ReaL` span multiple application domains. In particular, the suite includes:

- `DamEnv`, which exploits a mathematically validated model to manage a dam control system responsible for releasing the appropriate amount of water to meet residential demand;
- `ElevatorEnv`, which addresses a modified version of the elevator dispatching problem under dynamic request patterns;

Table 1: *Characteristics* and *RL Paradigms* covered by each environment provided by `Gym4ReaL`.

| | Characteristics | | | | | | RL Paradigms | | | | | |
|---|---|---|---|---|---|---|---|---|---|---|---|---|
| | Cont. States | Cont. Actions | Part. Observable | Part. Controllable | Non-Stationary | Visual Input | Freq. Adaptation | Hierarchical RL | Risk-Averse | Imitation Learning | Provably Efficient | Multi-Objective RL |
| DamEnv | ✓ | ✓ | | ✓ | | | | | | ✓ | | ✓ |
| ElevatorEnv | | | | ✓ | | | | | | | ✓ | |
| MicrogridEnv | ✓ | ✓ | | ✓ | | | ✓ | | | | | ✓ |
| RoboFeederEnv | ✓ | ✓ | | | | ✓ | | ✓ | | | | |
| TradingEnv | ✓ | | ✓ | ✓ | ✓ | | ✓ | | ✓ | | | |
| WDSEnv | ✓ | | | ✓ | | | | | | ✓ | | ✓ |

- `MicrogridEnv`, which adopts a digital twin framework to address the optimal energy management within a local microgrid, balancing supply, demand, and storage;
- `RoboFeederEnv`, which simulates in a virtual environment a robotic work cell tasked with isolating and picking small objects, including both picking and planning challenges;
- `TradingEnv`, which addresses the development of optimized trading strategies for the foreign exchange (Forex) market;
- `WDSEnv`, which employs a hydraulic analysis framework to model a municipal water distribution system, where the objective is to ensure a consistent supply to meet fluctuating residential demand.

Unlike prior works that address tasks in domain-specific contexts (see Appendix B), the contribution of `Gym4ReaL` is to provide a standardized implementation of these environments, fully compatible with the Gymnasium interface and grounded in realistic simulators and real-world datasets. Beyond supporting the training of agents tailored to these practical problems, `Gym4ReaL` is intentionally designed as a methodologically agnostic suite, enabling RL researchers to systematically evaluate and benchmark algorithms without requiring specialized domain knowledge.

**Scope and Contribution.** The primary goal of `Gym4ReaL` is not merely to supply environments for solving specific domain tasks, but rather to offer a curated suite of realistic environments encapsulating crucial challenges inherent to real-world applications for RL researchers, where they can validate new methods. Across the selected tasks, we emphasize both diversity and generalization in the goals and characteristics represented within the suite. A comprehensive summary of the suite's features is presented in Table 1. In particular, we distinguish between two key aspects: *Characteristics*, which refer to modeling properties specific to each environment, and *RL Paradigms*, which denote the classes of RL techniques that can be effectively tested and benchmarked within these environments beyond the classical RL approaches. While in this work we illustrate the utility of `Gym4ReaL` through benchmarking standard RL algorithms against expert-informed, rule-based baselines, the suite is expressly designed to accommodate a broader range of paradigms. For instance, the `DamEnv` task includes expert demonstrations that can be leveraged for imitation learning, inverse RL, or offline RL. Importantly, we include state-of-the-art algorithms to show that RL is well-suited for our environments: although their performance varies and is not optimal, they consistently outperform expert rule-based baselines. In this sense, our main goal is to provide a challenging and diverse environment suite, rather than an exhaustive algorithmic benchmark, which we leave for future work and for the community to extend. Eventually, `Gym4ReaL` offers a high degree of configurability. Users can customize input parameters and environmental dynamics to better reflect domain-specific requirements, thus extending the suite's usability to researchers from the respective application domains. Through this combination of realism, diversity, and flexibility, `Gym4ReaL` supports a wide spectrum of research efforts, from benchmarking general-purpose RL algorithms under realistic conditions to developing domain-specific controllers.

## 2 ENVIRONMENTS

This section introduces `Gym4ReaL` environments, describing each task objective and modeling. Test results derived by state-of-the-art RL algorithms are included and evaluated against expert-agreed rule-based baselines to establish that training on these environments is practical and yields sensible outcomes. Further details on environments and experiments are in Appendices E and F, while reproducibility instructions are provided in Appendix A.3.

### 2.1 DAMENV

`DamEnv` is designed to model the operation of a dam connected to a water reservoir. By providing the amount of water to be released as an action, the environment simulates changes in the water level, considering inflows, outflows, and other physical dynamics. The agent controlling the dam aims to plan the water release in order to satisfy the daily water demand while preventing the reservoir from exceeding its maximum capacity and causing overflows. Formally, the objective is:

$$\max \sum_{t=1}^{T} [r_d(a_t) + r_{\text{of}}(a_t) + r_{\text{st}}(a_t)], \tag{1}$$

where $r_d$ favors actions that meet daily demand, $r_{\text{of}}$ actions that prevent water overflows, and $r_{\text{st}}$ those that avoid starvation effects along the time horizon $T$. The daily control frequency adopted depends on the data granularity. Moreover, the available historical data derived from human-expert decisions allows for the development of imitation learning studies.

**Observation Space.** The observation space is composed as follows:

$$s_t = \left(l_t, \bar{d}_t, \cos(\varphi_t^y), \sin(\varphi_t^y)\right), \tag{2}$$

where $l_t$ is the water level at time $t$, $\bar{d}_t$ is the moving average of past water demands, and $\varphi_t^y \in [0, 2\pi]$ represents the angular position of the current time over the entire year, given by $\varphi^y = \frac{2\pi\tau_y}{T_y}$, where $\tau_y \in [0, T_y]$ is the current time in seconds and $T_y$ is the total number of seconds in a year.

**Action Space.** The action is a continuous variable $a_t \in \mathbb{R}^+$, representing the amount of water to release per unit of time.

**Reward Function.** The reward at time $t$ is $r_t = [r_d(a_t) + r_{\text{of}}(a_t) + r_{\text{st}}(a_t)] + \lambda_1 r_{\text{clip}}(a_t) + \lambda_2 r_w(a_t)$, where $r_d(a_t)$, $r_{\text{of}}(a_t)$ and $r_{\text{st}}(a_t)$ are the quantities in Equation 1, while $r_{\text{clip}}(a_t)$ and $r_w(a_t)$ are two terms designed to discourage actions beyond the physical constraints of the environment and to discourage water releases that are higher than the daily demand, respectively. The two positive hyperparameters $\lambda_1$ and $\lambda_2$ regulate the importance of these two additional penalty terms. The presence of multiple contrastive components enables the development of MORL paradigms.

**Benchmarking.** We employed an off-the-shelf implementation of the Proximal-Policy Optimization (PPO) (Schulman et al., 2017) algorithm as a benchmark state-of-the-art RL approach for the `DamEnv` task. We evaluated the trained agent against four rule-based baselines: the *Random* policy, which selects actions uniformly at random; the *Mean* policy, which selects the mean value of the action space; the *Max* policy, which selects the maximum value of the action space; and the *EAD* policy, which sets actions based on an exponential moving average of previous demands. The experiments conducted on 13 test episodes highlight the capability of the PPO agent to perform better than rule-based strategies. In particular, we can observe a better daily control of the PPO agent throughout one year, as shown in Figure 1a, and a larger average return with small variability, as highlighted in Figure 1b. Detailed results show that PPO avoids dam overflows much more effectively than the baselines, as detailed in the Appendix.

### 2.2 ELEVATORENV

`ElevatorEnv` is a simplified adaptation of the well-known elevator scheduling problem introduced by Crites & Barto (1995). Similarly to a subsequent work (Yuan et al., 2008), we design a discrete environment that simulates *peak-down traffic*, typical of scenarios such as office buildings at the end of a workday. In this environment, a single elevator serves a multi-floor building with $F$ floors and is tasked with transporting employees to the ground floor ($f = 0$). The episode unfolds over $T$ discrete

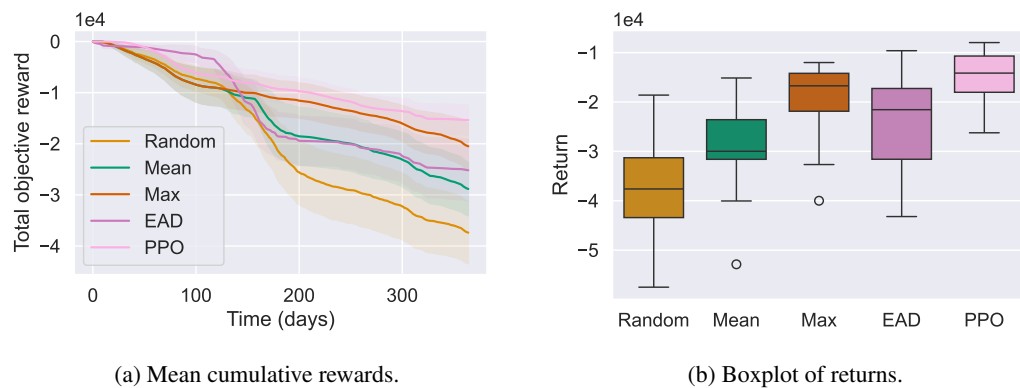

(a) Mean cumulative rewards.                    (b) Boxplot of returns.

Figure 1: Test performances with confidence intervals on `DamEnv`. Thirteen different episodes have been considered with a time horizon of one year.

time steps. At each floor $f \in \{1, \ldots, F\}$, new passengers arrive according to a *Poisson process* with rate $\lambda_f$.

Arriving passengers join a queue on their respective floor, provided the queue length is below a predefined threshold $W_{f,\max}$. Otherwise, they opt to take the stairs. The *goal* of the elevator controller is to minimize the cumulative *waiting time* of all transported passengers throughout the episode. This can be formalized as minimizing the cost:

$$\min \sum_{t=1}^{T} \Big( \sum_{f=1}^{F} w_{f,t} + c_t \Big), \tag{3}$$

where $w_{f,t}$ denotes the total waiting time of individuals at floor $f$ at time $t$. This setting defines a challenging *load management problem*, involving a trade-off between serving higher floors with longer queues and minimizing elevator travel time. Furthermore, the discrete and restrained formulation of `ElevatorEnv` facilitates the development of provably efficient RL methods, without losing the connection with the underlying real-world task.

**Observation Space.** The observation space is structured as follows:

$$s_t = (h_t, c_t, \mathbf{w}_t, \mathbf{k}_t), \tag{4}$$

where $h_t \in \{0, \ldots, H\}$ denotes the vertical position of the elevator within the building at time $t$, being $H$ the maximum reachable height, $c_t \in \{0, \ldots, C_{\max}\}$ indicates the current load of the elevator, in number of passengers, up to the maximum capacity $C_{\max}$, and $\mathbf{w}_t \in \mathbb{N}^F$ and $\mathbf{k}_t \in \mathbb{N}^F$ represent the actual number of people waiting in the queue and the new arrivals at each floor.

**Action Space.** The action space is defined by the discrete action variable $a_t \in \{u, d, o\}$ which indicates whether the elevator has to move upwards ($u$), move downwards ($d$), or stay stationary and open ($o$) the doors. Actions are mutually exclusive and applied at each time step $t$.

**Reward Function.** The instantaneous reward is $r_t = -(\sum_f w_{f,t} + c_t) + \mathbb{1}_{\{c_t=0\}} \beta \, c_{t-1}$, i.e., at each step $t$ we penalize the presence of individuals, either waiting in queues ($w_{f,t}$) or inside the elevator ($c_t$), as in Equation equation 4. In addition, we grant a positive reward when passengers are successfully delivered to the ground floor, i.e., when the elevator becomes empty. The positive hyperparameter $\beta > 0$ controls the reward magnitude for offloading $c_{t-1}$ passengers.

**Benchmarking.** For the `ElevatorEnv` task, we adopt two well-known tabular RL algorithms: Q-Learning (Watkins & Dayan, 1992) and SARSA (Sutton & Barto, 2018). Such methods are evaluated against different rule-based strategies, i.e., the *Random* policy, and the *Longest-First* (LF) and the *Shortest-First* (SF) policies, which prioritize the floor with a higher or lower number of waiting people, respectively. As shown in Figure 2a, both RL algorithms consistently outperform the other rule-based solutions, considerably reducing the global waiting time. In particular, as reported in Figure 2b, Q-Learning shows higher performance than SARSA, which, due to its inherent nature, tends to play more conservative actions.

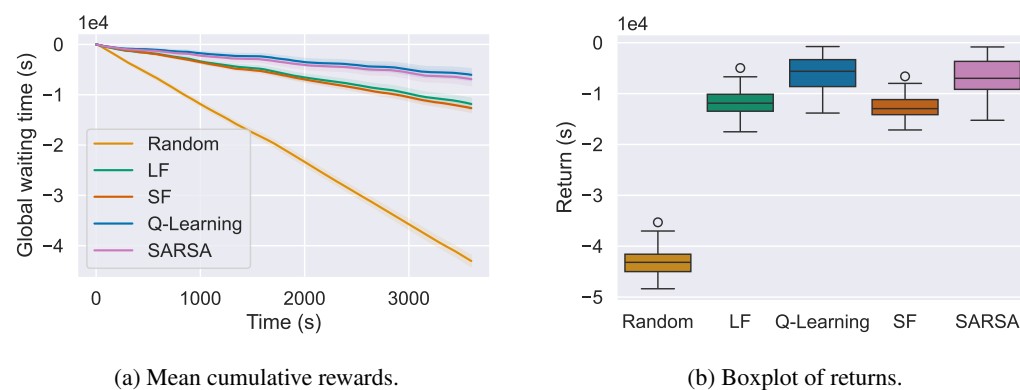

(a) Mean cumulative rewards.

(b) Boxplot of returns.

Figure 2: Performance of baselines in terms of mean cumulative reward (a) and average return (b) on `ElevatorEnv`. Results collected over 30 different episodes.

## 2.3 MICROGRIDENV

`MicrogridEnv` simulates the operation of a microgrid within the context of electrical power systems. Microgrids are decentralized components of the main power grid that can function either in synchronization or in islanded mode. In this scenario, the control point is placed on the battery component, which must find the best strategy to manage the accumulated energy over time optimally. Formally, the controller wants to maximize its total profit over a time horizon of $T$. Hence, the objective is:

$$\max \sum_{t=1}^{T} [r_{\text{trad}}(a_t) + r_{\text{deg}}(a_t)], \tag{5}$$

where $r_{\text{trad}}(a_t) \in \mathbb{R}$ is the reward/cost gained from the exchanges of energy with the market, and $r_{\text{deg}}(a_t) < 0$ is the cost due to battery degradation. The benchmark leverages real-world datasets, as detailed in the Appendix, and the battery behavior is modeled using a digital twin of a BESS (Salaorni et al., 2025). Each episode is formulated as an infinite-horizon problem and terminates either when the dataset is exhausted or the battery reaches its end-of-life condition. Moreover, the presence of energy market trends allows the usage of `MicrogridEnv` for frequency adaptation analysis.

**Observation Space.** The observation space comprises variables regarding the internal state of the system and uncontrollable signals received from the environment. Formally:

$$s_t = \big(\sigma_t, K_t, \widehat{P}_{D,t}, \widehat{P}_{G,t}, p_t^{\text{buy}}, p_t^{\text{sell}}, \cos(\varphi_t^d), \sin(\varphi_t^d), \cos(\varphi_t^y), \sin(\varphi_t^y)\big), \tag{6}$$

where $\sigma_t$ is the storage state of charge, $K_t$ is the battery temperature, $\widehat{P}_{D,t}$ is the estimate of energy demand $P_{D,t}$, $\widehat{P}_{G,t}$ is the estimate of energy generation $P_{G,t}$, $p_t^{\text{buy}}$ and $p_t^{\text{sell}}$ are the buying and selling energy market prices, respectively, $\varphi_t^d \in [0, 2\pi]$ is the angular position of the clock in a day, and $\varphi_t^y \in [0, 2\pi]$ is the angular position of the time over the entire year.

**Action Space.** The action space is determined by the continuous action variable $a_t \in [0, 1]$, representing the proportion of energy to *dispatch* (*take*) to (from) the BESS. The action operates with the net power computed as $P_{N,t} = P_{G,t} - P_{D,t}$. If $P_{N,t} > 0$, it regulates the proportion of energy used to charge the battery or sold to the main grid. Conversely, if $P_{N,t} < 0$, the action balances the proportion of energy taken from the energy storage or bought from the market.

**Reward Function.** The instantaneous reward is $r_t = [r_{\text{trad}}(a_t) + r_{\text{deg}}(a_t)] + \lambda r_{\text{clip}}(a_t)$, where $r_{\text{clip}}(a_t)$ is a penalty that discourages actions that do not respect physical constraints, weighted by the hyperparameter $\lambda$. The first two elements, instead, are the same components of the objective function in Equation equation 5, whose contrastive optimization enables multi-objective RL approaches.

**Benchmarking.** For the `MicrogridEnv`, we compare an RL agent trained with PPO against several rule-based policies: the *Random* policy; the *Only-market* (OM) policy, which forces the interaction with the grid without using the battery; the *Battery-first* (BF) policy, which fosters the battery usage; and the *50-50* policy, which adopts a behavior in the middle between OM and BF. Figure 3a shows that, during testing, PPO achieves higher profit than rule-based strategies. However,

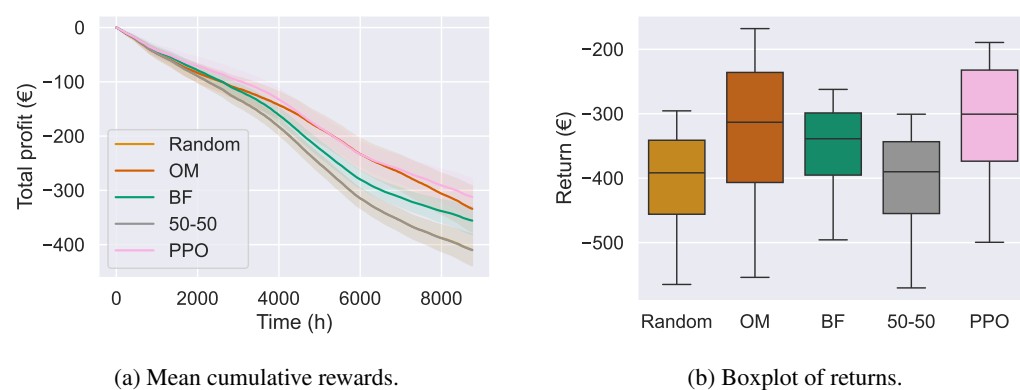

(a) Mean cumulative rewards.          (b) Boxplot of returns.

Figure 3: Performance of baselines in terms of mean cumulative reward (a) and average return (b) on `MicrogridEnv`. Results have been collected over 28 different episodes.

as reported in Figure 3b, PPO has a large variance, suggesting the need for novel RL algorithms to achieve more consistent behavior.

## 2.4 ROBOFEEDERENV

`RoboFeederEnv` is a collection of environments designed to pick small objects from a workspace area with a 6-degree-of-freedom (6-DOF) robotic arm. This task involves two primary challenges: determining the *picking order* of the objects and identifying the precise *grasping point* on each object for successful pickup and placement. To closely mimic the behavior of the commercial robotic system, a simulation emphasizing contact interactions is conducted using MuJoCo. This environment supports goal-oriented training, enabling the robot to learn how to identify the appropriate grasping points and, more broadly, to determine the most efficient order of picking. Unlike most robotic simulators, `RoboFeederEnv` is uniquely tailored to operate at the trajectory planning level rather than through low-level joint control, which is more realistic in industrial applications, given the impossibility of accessing and modifying proprietary kinematic controllers.

Due to the hierarchical nature of the problem, we split the setting into two underlying environments: `RoboFeeder-picking` and `RoboFeeder-planning`.

### 2.4.1 ROBOFEEDER-PICKING

`Gym4ReaL` includes two types of picking environments of increasing difficulty:

- `picking-v0`: a simpler environment where the top-down image is pre-processed by cropping around detected objects, reducing the complexity of the visual input, thus of the observation space;
- `picking-v1`: a more challenging environment where the observation is the full camera image.

**Observation Space.** The observation is defined by the visual input $s_t = \mathbf{X}_t \in \mathbb{R}^{H \times W \times C}$, where each image $\mathbf{X}_t$ is represented by a tensor of height $H$, width $W$, and channel $C$, and is captured by a camera positioned on top of the working area. Within the `picking-v0` environment, the image tensor is restricted to $\widehat{\mathbf{X}}_t \in \mathbb{R}^{\widehat{H} \times \widehat{W} \times C}$, with $\widehat{H}$ and $\widehat{W}$ cropped image dimensions.

**Action Space.** The action space is determined by the continuous action $a_t = (x_t, y_t)$, where $(x_t, y_t)$ are relative coordinates within the segmented image, corresponding to the target grasping point.

**Reward Function.** The reward function is designed to foster successful object picking while penalizing unfeasible or suboptimal actions. Formally, the instantaneous reward is $r_t = 1$ if the object is correctly picked up, $r_t = -1$ if the action is unfeasible, or $r_t = -1 + r_{d,t} + r_{\theta,t}$ otherwise, where $r_{d,t}$ is a distance-based shaping term that rewards proximity of the end-effector to the object, and $r_{\theta,t}$ is a rotation-based shaping term that incentivizes alignment with the desired grasping orientation.

**Benchmarking.** We evaluate the performance of a trained PPO agent against a fixed action rule-based strategy on the `picking-v0` environment. The task involves objects uniformly distributed within the workspace, requiring non-trivial generalization capabilities. Figures 4a and 4b report how the

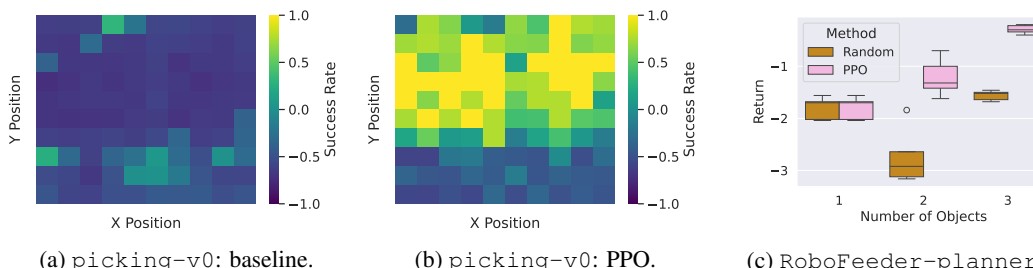

(a) `picking-v0`: baseline.  (b) `picking-v0`: PPO.  (c) `RoboFeeder-planner`.

Figure 4: Heatmap of the success rate of picking tasks across the entire workspace with baseline (a) and PPO (b) (the higher, the better). Comparison between *Random* policy and PPO within the planning problem (c) (average return over $50$ episodes and $5$ different random seeds).

baseline exhibits consistently poor performance, while the PPO agent achieves higher and more evenly distributed success rates, highlighting its capability to learn an effective picking strategy.

### 2.4.2 ROBOFEEDER-PLANNING

The `RoboFeeder-planning` is an environment aiming to decide the order to follow for picking the objects in the work area. It is a high-level task w.r.t. `RoboFeeder-picking`, not involving the direct control of the robot, but only concerning the optimal picking schedule.

**Observation Space.** The observation space is defined by the vector of visual input $s_t = [\mathbf{X}_{1,t}, \ldots, \mathbf{X}_{N,t}]$, with $\mathbf{X}_{i,t} \in \mathbb{R}^{H \times W \times C}$, where $N$ is the maximum number of images that can be processed and $\mathbf{X}_{i,t}$ is an image defined as in the `picking-v0` task. Each of the $N$ image patches corresponds to a cropped and scaled region of a detected object.

**Action Space.** The action space is determined by the discrete action $a_t \in \{0, 1, \ldots, N\}$, selecting the image from $1$ to $N$ containing the object to pick. Action $0$, instead, is a special *idle* action that can be chosen when no graspable objects are available. This formulation enables continuous deployment since the robot can remain idle while waiting for the arrival of new objects.

**Reward Function.** The immediate reward is $r_t = 1$, if the selected object is correctly picked, $r_t = -1$ if it is not picked, and $r_t = -\sum_{i=1}^{M} \mathbb{1}_{\{\text{obj}_i \text{ not picked but graspable}\}}$ if the agent plays the *idle* action $a_t = 0$ while graspable objects are present, with $M$ being the currently available objects.

**Benchmarking.** In Figure 4c, we compare the efficiency of a trained PPO agent against a *Random* strategy. Results highlight the agent's capability to determine an optimal picking schedule by distinguishing objects placed in a favorable position to be picked up. Moreover, as the number of objects increases, the gap between the average return of PPO and the baseline increases too.

### 2.5 TRADINGENV

`TradingEnv` provides a simulated market environment, trained with historical foreign exchange (Forex) data relative to the EUR/USD currency pair, where the objective is to learn a profitable intraday strategy. The problem is framed as episodic: each episode starts at 8:00 EST and ends at 18:00 EST when the position must be closed. At each step, based on its expectations, the agent can open a *long* position (i.e., buy a fixed amount of the asset), remain *flat* (i.e., take no action), or open a *short* position (i.e., short sell a fixed amount of the asset). Typical baselines include passive strategies, such as *Buy&Hold* (B&H) and *Sell&Hold* (S&H), which consist of maintaining fixed positions.

Trading tasks are typically subjected to several challenges. For example, the state has to be carefully designed to deal with the low signal-to-noise ratio, and it is typically large-dimensional, including past prices and temporal information. Moreover, the environment is partially observable, and financial markets are non-stationary. Another relevant aspect is the calibration of the trading frequency, considering the amount of noise and transaction costs. In addition, risk-aversion approaches can be of interest, considering not only the profit-and-loss (P&L) but also the variance among episodes.

**Observation Space.** The observation space is composed of two components: *market state* and *agent state*. The *market state* includes calendar features and recent price variations, namely the last 60 delta

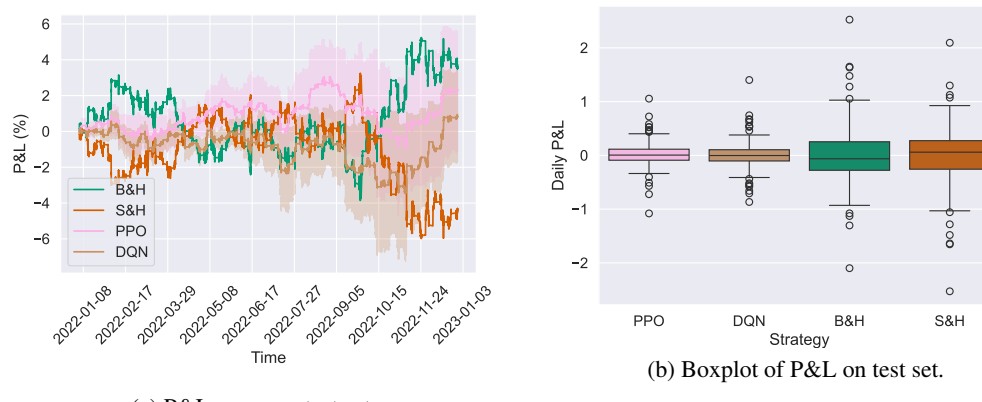

(a) P&L curve on test set.

(b) Boxplot of P&L on test set.

Figure 5: Performances of PPO and DQN against baselines B&H and S&H on Test (a) Daily Performance on Test (b) on `TradingEnv`. Mean and Confidence Intervals computed using 6 seeds.

mid-prices, where a delta mid-price is defined as $d_{k,t} = \frac{p_{t-k} - p_{t-k-1}}{p_{t-k-1}}$, with $k \in \{0, \ldots, 59\}$. The *agent state* component, on the contrary, includes the current position $z_t$, that is, the action that was previously played. Formally, the state in this setting is:

$$s_t = \left( \mathbf{d}_t, \cos(\varphi_t^{day}), \sin(\varphi_t^{day}), z_t \right), \tag{7}$$

where $\mathbf{d}_t = [d_{0,t}, \ldots, d_{59,t}]$ is the vector of the last 60 delta mid prices at time $t$, $\varphi_t^{day} \in [0, 2\pi]$ is the angular position of the current time over the trading period, and $z_t = a_{t-1}$ is the agent position.

**Action Space.** The action space is determined by a discrete variable $a_t \in \{s, f, l\}$, where $s$ (*short*) indicates that the agent is betting against EUR, supposing a decline in the value relative to USD; $f$ (*flat*) indicates no market exposition; and $l$ (*long*) means that the agent expects that the relative EUR value will increase. Each action refers to a fixed amount of capital $C$ to trade.

**Reward Function.** The immediate reward at time $t$ is the signal $r_t = a_{t-1}(p_t - p_{t-1}) - \lambda |a_t - z_t|$, where the first term is related to the P&L obtained from a price change, and the second component regards the commissions paid when the agent changes its position, being $\lambda$, a constant transaction fee.

**Benchmarking.** We trained agents using off-the-shelf implementations of PPO and Deep Q-Network (DQN) (Mnih et al., 2015) on `TradingEnv`. Their performance against common passive baselines, B&H and S&H, are evaluated on a test year (Figure 5a). As expected, neither PPO nor DQN is able to consistently outperform the baselines, due to the complexity of the problem. However, RL remains a valid candidate to tackle trading tasks, as it significantly reduces the daily variability of the P&L (Figure 5b).

## 2.6 WATERDISTRIBUTIONSYSTEMENV

`WaterDistributionSystemEnv` simulates the evolution of a hydraulic network in charge of dispatching water across a residential town. A network is composed of different entities, such as storage tanks, pumps, pipes, junctions, and reservoirs, and the main objective of the system is the safety of the network. To achieve such a goal, we have to ensure optimal management of hydraulic pumps, which are in charge of deciding how much water should be collected from reservoirs and dispatched to the network. The pumps' controller must guarantee network resilience by maximizing the demand satisfaction ratio (DSR) while minimizing the risk of overflow. Formally, the objective is

$$\max \sum_{t=1}^{T} [r_{\text{DSR}}(a_t) + r_{\text{of}}(a_t)], \tag{8}$$

where $r_{\text{DSR}}(a_t) \in [0, 1]$ is the ratio between the supplied demand on the expected demand at time $t$, and $r_{\text{of}}(a_t) \in [0, 1]$ is a normalized penalty associated with the tanks' overflow risk.

The environment leverages the hydraulic analysis framework Epanet (Rossman, 2000), which provides the mathematical solver for water network evolution, and realistic datasets of demand profiles.

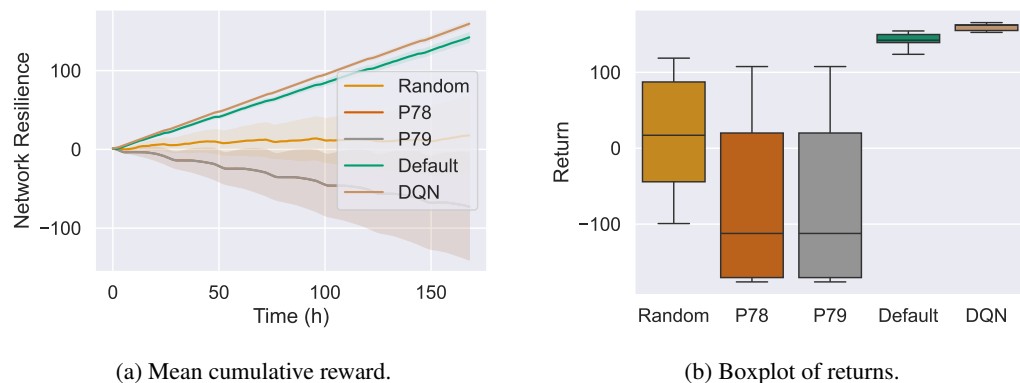

(a) Mean cumulative reward.

(b) Boxplot of returns.

Figure 6: Performance of baselines in terms of mean cumulative resilience (a) and average return (b) on `WDSEnv`. Results have been collected over 20 different episodes.

Therefore, `WDSEnv` may also be suitable to test imitation learning methods, having at disposal an expert policy from the *.inp* configuration file of networks read by Epanet.

**Observation Space.** The observation space includes the internal state of the network and an estimation of the global demand profile that the system is asked to deal with. Formally:

$$s_t = \left( \mathbf{h}_t, \mathbf{p}_t, \widehat{d}_t, \cos(\varphi_t^d), \sin(\varphi_t^d) \right), \tag{9}$$

where $\mathbf{h}_t \in \mathbb{R}^L$ is the vector of $L$ tank levels at time $t$, $\mathbf{p}_t \in \mathbb{R}^J$ is the vector of $J$ junction pressures at time $t$, $\widehat{d}_t$ is the estimated total demand at time $t$, and $\varphi_t^d \in [0, 2\pi]$ is the angular position of the clock in a day. Finally, although all tanks must be monitored, we can reduce the dimensionality of the observation space by considering only junctions placed in strategic positions.

**Action Space.** The discrete action variable $a_t \in \mathbb{N}$ can assume values in $\{0, \ldots, 2^P - 1\}$, with $P$ number of pumps within the system. The action determines the combination of open/closed pumps.

**Reward Function.** The instantaneous reward given by the environment is $r_t = r_{\text{DSR},t}(a_t) + r_{\text{of},t}(a_t)$, where the terms are those described in the objective function in Equation equation 8.

**Benchmarking.** The `WDSEnv` is benchmarked adopting DQN, which is compared with different rule-based baselines: the *Random* policy, *P78* and *P79* policies, which act by keeping active only the relative pump (namely P78 or P79, respectively), and the *Default* policy, which executes the default control rules contained within the *.inp* configuration file of the network, changing the control action depending on the current tank level. As depicted in Figure 6a, DQN achieves a higher level of resilience with respect to other baselines. Moreover, Figure 6b shows that it has a more consistent behavior and low variance, a crucial characteristic for the resilience and safety of the water network.

## 3 DISCUSSION AND CONCLUSIONS

In this work, we presented `Gym4ReaL`, a reference environment suite developed in collaborations with industries and research centers and designed to realistically model several real-world environments, built on research-grade simulators and real-world datasets. Unlike standard RL suites, such as Gymnasium and MuJoCo, `Gym4ReaL` represents a novel library that allows for evaluating new RL methods in realistic applications. Notably, the `Gym4ReaL` suite includes environments designed to capture common real-world challenges, such as limited data availability, realistic assumptions about physical process dynamics, and constrained exploration, fostering research toward broader adoption of RL methods in practical applications. Indeed, the variety of tasks and challenges tackled with the presented suite offers the opportunity to address multiple *RL Paradigms* across environments with different *Characteristics*, as highlighted in Table 1. Given the standardized and flexible interface offered by our suite, a broader range of real-world problems could be easily integrated into our framework. We believe that a collective effort from the RL community can significantly advance the development of realistic, impactful benchmarks. Hence, we encourage researchers and practitioners to explore, contribute to, and adopt `Gym4ReaL` to evaluate RL algorithms in real-world scenarios.

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
