# OpenReview forum: "Gym4ReaL: Towards Real-World Reference Environments for Reinforcement Learning"
_ICLR.cc/2026/Conference — ICLR 2026 Conference Withdrawn Submission_

### Official Review · Reviewer_WwGz · 2025-10-20

**Soundness:** 1
**Presentation:** 3
**Contribution:** 1
**Rating:** 2
**Confidence:** 4

**Summary:**

This paper introduces Gym4ReaL, a collection of reinforcement learning (RL) environments intended to bridge the gap between simulation and real-world domains. The suite includes tasks such as elevator dispatching, dam control, and trading, each implemented with a standardized Gymnasium interface. The authors claim these environments capture real-world complexities that are underrepresented in existing benchmarks and aim to promote more realistic RL research.

**Strengths:**

- The paper provides clear and detailed descriptions of each environment, including reward design, observation spaces, and benchmark setup.
- The paper deploys PPO, DQN, and tabular methods as appropriate in the environments.
- Compatibility with the Gym interface supports potential integration into existing RL pipelines.
- The motivation to broaden RL benchmarks beyond classical robotics and control is, in principle, worthwhile.

**Weaknesses:**

- The abstract claims that for RL “success has not been translated directly to real-world domains,” yet many prior works (e.g., HumanoidBench [3], Isaac Sim, and many other sim-to-real studies) have demonstrated effective real-world transfer of RL policies. The paper specifies some real-world complexities they claim remain unaddressed but does not sufficiently motivate how the Gym4ReaL environment selections concretely capture them better than prior work or provide evidence that prior work has not addressed these same issues. There is no related work section, making the overlap and positioning relative to prior RL benchmarks very unclear.

- As a follow-on to the former point, the selected tasks (dam control, elevator dispatching, trading) are unconventional but not clearly motivated. The authors assert that these environments cover a wider range of paradigms, but Table 1 lacks quantitative or comparative evidence to show greater diversity relative to existing benchmarks. The selection appears arbitrary and driven by engineering convenience rather than by a principled analysis of benchmark gaps.

- Section 2 provides good implementation detail but evaluates only a few RL algorithms: PPO for the dam task and tabular Sarsa/Q-learning for the elevator task (adapted from a 1995 study [1]). Other environments use only PPO and DQN. This narrow evaluation scope is insufficient to establish benchmarking value if we compare to prior work. By contrast, major suites such as DeepMind Control Suite [2] and HumanoidBench [3] compare a wide range of algorithms (e.g., A3C, D4PG, SAC, TD-MPC2, DreamerV3) across pixel- and state-based settings.

- The overall contribution is therefore not well differentiated from prior work. The paper does not make a compelling case for how these environments fill known gaps in benchmark coverage or offer unique real-world relevance. Without this, the contribution reads as “Gym interfaces for niche environments” rather than a meaningful community benchmark.

- The appendix is cited repeatedly but not provided, which leaves out details that are seemingly missing such as Gym4ReaL’s contrast with “domain-specific contexts.” Maybe there were more details provided there in terms of comparing and contrasting to prior benchmarks, but those details should be front and center as motivation rather than relegated to an appendix.

References:\
[1] Crites & Barto, Improving Elevator Performance Using Reinforcement Learning, NIPS 1995.

[2] Tassa et al., DeepMind Control Suite, arXiv:1801.00690.

[3] Sferrazza et al., HumanoidBench, arXiv:2403.10506.

**Questions:**

- What specific “real-world complexities” do these environments model that are absent from existing simulation benchmarks, and where is the supporting evidence for this claim (e.g., through a related work or literature review)?
- Why were only PPO and DQN (and tabular methods) selected for evaluation? Would the results hold for other modern RL algorithms such as SAC, TD3, or DreamerV3?
- If results were included for more modern methods, would they prove to be overkill for the complexity of the environments proposed as part of Gym4ReaL, or would they fail to work due to overtuning for continuous control in high-dimensional robotics tasks? In other words, do these environments reveal genuine algorithmic limitations that existing benchmarks fail to expose?

---

### Official Review · Reviewer_ruwq · 2025-10-27

**Soundness:** 2
**Presentation:** 3
**Contribution:** 1
**Rating:** 0
**Confidence:** 5

**Summary:**

This paper introduces Gym4ReaL, a set of 6 RL environments compatible with Gymnasium. These environments are designed to emulate real-world scenarios in which RL could be applied.

**Strengths:**

The paper is well-written and easy to understand. It tackles an important problem of applying RL to real world scenarios.

**Weaknesses:**

1. The scope of the library (and consequently the paper) is very limited - only 6 environments, none of which are particularly innovative
2. Some (but not all) of the environments are quite simple, consisting of a few python files
3. The code itself is not particularly optimized, nor is the quality particularly high. The environments are mostly compatible with Gymnasium, but not following standard conventions and best practices (for example: TradingEnv using a custom argument in the reset method, instead of using `options` which are meant exactly for that purpose)



Overall, my biggest problem with this work is that its scope is extremely limited, and I fail to see a contribution to RL more broadly. The biggest challenges in applying RL to real world problems are (a) the need to build custom efficient simulators, and (b) potential sim-to-real challenges (aside from the usual limitations of RL algorithms). This work introduces six very specific environments that do not stand out

**Questions:**

Unfortunately I fail to see anything that would convince me to change my recommendation, as the paper's scope is way too small for a publication. As it stands, it is a set of six very specific environments with various levels of polish.

However, I do want to commend the authors for their work, and recommend working towards a more general, unifying framework for real-life RL scenarios. A framework that can be demonstrably applied to a wide range of real-life problems would be highly valuable - but also challenging to create.

---

### Official Review · Reviewer_MgG7 · 2025-11-01

**Soundness:** 2
**Presentation:** 2
**Contribution:** 2
**Rating:** 6
**Confidence:** 4

**Summary:**

Gym4ReaL is a first attempt to bridge the gap between synthetic benchmarks and real‑world applications in reinforcement learning.
Its selling point lies mainly in a curated set of realistic environments derived from industry collaborations and its open‑source nature.

However, several of the provided environments remain simplified in ways that limit their real-world fidelity. As a result, their current form still feels closer to conceptual or synthetic benchmarks than to truly deployed scenarios. Substantial enhancement of environment realism and baseline complexity would be needed for the suite to gain broad traction and fuel robust research in real-world RL.

The proposed environments would need substantial improvements to gain traction in the community and hope for further extensions in the future, which would be nice to see.

**Strengths:**

The paper addresses a critical gap in RL benchmarking by proposing Gym4ReaL, a suite of realistic environments inspired by real-world applications. Its key strengths include:
- Introducing diverse, industry-backed environments that go beyond synthetic simulations.
- Offering a unified, Gym-compatible interface that encourages the community to experiment with more realistic tasks.
- Highlighting practical challenges like partial observability and constrained exploration.
- Laying the groundwork for research in multi-objective, hierarchical, and safety-focused RL.

**Weaknesses:**

While Gym4ReaL is a promising step toward realistic RL benchmarking, the current suite falls short of its stated ambitions due to several key limitations:

1. Reward Design and Multi-Objective Considerations
Many tasks involve inherently multi-objective trade-offs (e.g., safety vs. performance), yet reward functions are fixed and not exposed for user customization. This restricts exploration and does not reflect the complexity of real-world decision-making.

2. Evaluation Methodology
The evaluation strategy is limited, often relying on training data and basic baselines, with little consideration of domain-informed or state-of-the-art alternatives. As a result, comparative performance claims lack robustness and practical relevance.

3. Environment Realism
Several environments make simplifying assumptions that diminish their real-world fidelity:
 - DamEnv: DamEnv: The environment models dam operations but lacks realistic control dynamics and failure conditions (e.g., overflows) beyond a fixed reward structure. Baselines such as constant max or random release do not reflect operational constraints or expert-driven water resource policies.
- ElevatorEnv: The model assumes perfect observability of queue lengths and restricts the task to downward traffic only, ignoring multi-directional flow and passenger behavior typical in real elevator systems. The arrival process is synthetic (Poisson), whereas real usage data could be incorporated for greater fidelity.
- MicrogridEnv: The environment lacks clear specification of the energy source (e.g., solar, wind), and the use of “negative profit” is ambiguous. Market interactions are modeled in a simplified way, without accounting for key external drivers such as day-ahead prices, weather uncertainty, or reserve requirements.
- RoboFeeder: (Picking and Planning): The picking environment lacks meaningful baselines and does not provide visual examples of the workspace or robot interactions. In the planning environment, comparing PPO only against a random baseline does not convincingly demonstrate learning for even small object counts.
- TradingEnv: The state space includes only price deltas and encodes neither market microstructure nor economic indicators. Fixed trading volume and simplistic long/flat/short actions neglect key dimensions such as position sizing, leverage, and risk management.
- WDSEnv: The water network appears symmetric across two pumps (P78 and P79), leading to identical outcomes in plots, likely due to model oversimplification. Baselines use naive pumping strategies that do not reflect standard water distribution operations or control logic.

Overall, the environments remain closer to synthetic benchmarks than to the real-world scenarios they aim to emulate, requiring substantial enhancement for their intended role in advancing practical RL research.

**Questions:**

-  How do you plan to increase the realism and complexity of the current environments in future versions (e.g., richer dynamics or multi-agent settings)?
- Can users customize or extend the reward functions, for example to support multi-objective or safety constraints, within the provided environments?
- Why were simple rule-based policies used as baselines? Do you plan to include stronger or domain-specific baselines?
- Will additional real datasets or more varied domains (e.g., healthcare, logistics) be integrated into future releases?
- How do you ensure reproducibility in environments that rely on real-world datasets or stochastic simulations?

**Details Of Ethics Concerns:**

No ethical concerns

---

### Note · Authors · 2025-11-17

**Comment:**

We thank the Reviewers for the detailed feedback, we decided to withdraw the work in order to improve it following Reviewers comments.

**Withdrawal Confirmation:**

I have read and agree with the venue's withdrawal policy on behalf of myself and my co-authors.